# Evidential Learning-based Certainty Estimation for Robust Dense Feature Matching

**Lile Cai    Chuan-Sheng Foo    Xun Xu    Zaiwang Gu    Jun Cheng    Xulei Yang**

Institute for Infocomm Research (I²R), A*STAR, Singapore

{`caill,foo_chuan_sheng,xu_xun,gu_zaiwang,cheng_jun,yang_xulei`}@i2r.a-star.edu.sg

## Abstract

Dense feature matching methods aim to estimate a dense correspondence field between images. Inaccurate correspondence can occur due to the presence of unmatchable region, necessitating the need for certainty measurement. This is typically addressed by training a binary classifier to decide whether each predicted correspondence is reliable. However, deep neural network-based classifiers can be vulnerable to image corruptions or perturbations, making it difficult to obtain reliable matching pairs in corrupted scenario. In this work, we propose an evidential deep learning framework to enhance the robustness of dense matching against corruptions. We modify the certainty prediction branch in dense matching models to generate appropriate belief masses and compute the certainty score by taking expectation over the resulting Dirichlet distribution. We evaluate our method on a wide range of benchmarks and show that our method leads to improved robustness against common corruptions and adversarial attacks, achieving up to 10.1% improvement under severe corruptions.

## 1 Introduction

Feature matching is an important task in computer vision. It aims to find a set of matching pixel pairs between two images, based on which two-view geometry, *e.g.*, homography and relative camera pose, can be derived. There are three paradigms to perform feature matching: detector-based (Lowe, 2004; DeTone et al., 2018; Sarlin et al., 2020), detector-free (Sun et al., 2021) and dense matching (Truong et al., 2023; Edstedt et al., 2023; 2024). It has been demonstrated that dense matching methods obtain superior accuracy compared to others on a wide range of geometry estimation benchmarks (Edstedt et al., 2023; 2024). We focus on the dense matching paradigm in this paper.

Dense feature matching estimates a 2D dense correspondence field for the input image pair. It typically works in a coarse-to-fine manner, where matches are first predicted at a coarse scale and subsequently refined at finer scales. As it cannot be guaranteed that every pixel in the two images are matchable, it is necessary to measure the certainty for each predicted correspondence and only use reliable matches for downstream geometry estimation. This is usually formulated as a binary classification problem, where pixels deemed to have correspondence in the other image (based on ground truth geometry information) are classified into 1 while unmatchable pixels are classified into 0 (Melekhov et al., 2019; Edstedt et al., 2023; 2024). The classification probability for class 1 is thus indicating matchability and used as certainty score. When applied to downstream geometry estimation tasks, a balanced sampling step (Edstedt et al., 2023) is adopted to sample a sparse set of matches from the dense matching results, where matches with larger certainty score have higher chance to be selected.

State-of-the-art (SotA) dense feature matching methods are built on deep neural networks, which have been shown to suffer severe performance loss on images under adversarial attacks (Goodfellow et al., 2014) or common corruptions (Hendrycks & Dietterich, 2019; Michaelis et al., 2019). Moreover, 3D datasets used to train feature matching models are usually much smaller in size than 2D datasets, which may limit the model robustness to testing data under distribution shift. Previous benchmarking datasets (Mishkin et al., 2015; Howard et al., 2022) do not consider images under common corruptions or adversarial attacks, both of which are relevant – the former can happen in

real deployment environment, and the latter provides a worst-case analysis of model robustness. Our work fills this gap. As shown in Fig. 1, the SotA dense matcher RoMa fails to provide reliable certainty estimation in matchable regions when the input images are corrupted by Gaussian noise or adversarial attacks, resulting in large pose estimation error. In particular, RoMa tends to produce very low certainty value (e.g., below 0.05) for matchable regions on corrupted images, *i.e.*, the model over-confidently classifies the pixels as unmatchable. The topic of improving the robustness of feature matching methods under image corruptions remains largely unexplored in the literature.

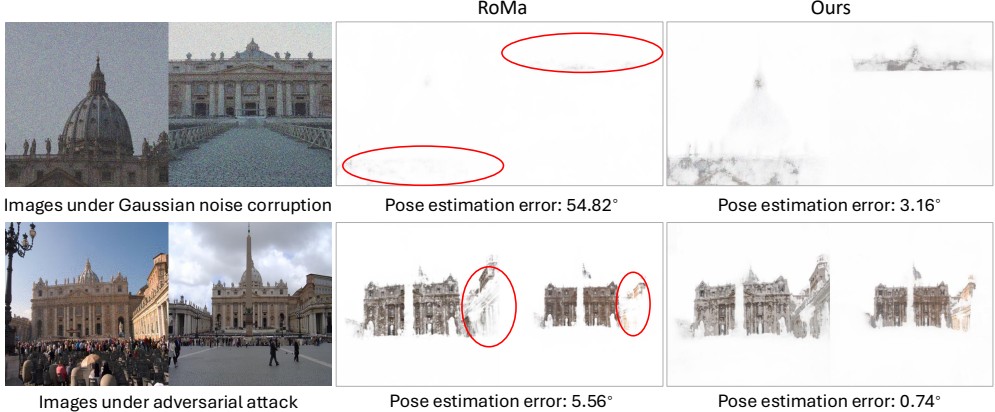

Figure 1: SotA dense matcher RoMa fails to provide reliable certainty estimation in matchable regions (indicated by red circles) when the input images are corrupted by Gaussian noise or adversarial attacks, resulting in large pose estimation error. Our method is able to obtain reliable feature matches under image corruptions and adversarial attacks. The matching results are visualized by taking the pixel values from the other image using the estimated correspondences, and weighting by the estimated certainty.

Having identified unreliable certainty estimation, or over-confidence, as a bottleneck to robust feature matching, we hypothesize that improved uncertainty estimation will lead to improved feature matching performance. To this end, we explore evidential deep learning (EDL) for certainty estimation in dense feature matching methods. Instead of providing a direct pointwise estimation of probability, EDL models second-order probabilities under the framework of subjective logic theory (Josang, 2016). It has been shown to provide improved uncertainty estimation for detecting out-of-distribution samples and mitigate the over-confidence problem (Sensoy et al., 2018).

Our contributions can be summarized as below:

- We propose an evidential learning framework specifically designed for certainty estimation in dense feature matching. To the best of our knowledge, this is the first application of EDL in feature matching tasks.
- We propose to evaluate the robustness of feature matching methods under common image corruptions and adversarial attacks, which has not been studied in previous work.
- We evaluate our method on both corrupted data and clean data, and show that our method leads to improved robustness against common corruptions and adversarial attacks without sacrificing the performance on clean data.

## 2 RELATED WORK

### 2.1 FEATURE MATCHING

Feature matching is a long-standing task in computer vision. The detector-based approach first performs sparse keypoint detection and descriptor extraction, followed by mutual nearest neighbour matching (Lowe, 2004; DeTone et al., 2018) or learning-based matching (Sarlin et al., 2020). Detector-free approach removes the keypoint detection step and calculates the pairwise similarity score uniformly over the image grid. Matches are then extracted from the similarity matrix using

mutual-nearest neighbour or optimal transport (Sun et al., 2021). The dense matching approach (Truong et al., 2023; Edstedt et al., 2023; 2024) estimates a 2D flow field, aiming to find all matchable pixel pairs between the two images.

## 2.2 CERTAINTY ESTIMATION IN DENSE MATCHING

The flow field predicted by dense matching methods may come with inaccurate correspondences in unmatchable regions, necessitating a measure of certainty for each predicted correspondence. DGC-Net (Melekhov et al., 2019) formulates it as a binary classification problem to identify out-of-view pixels. DKM (Edstedt et al., 2023) and RoMa (Edstedt et al., 2024) also train a binary classifier where pixels corresponding to the same 3D points are classified as reliable matches. PDC-Net (Truong et al., 2023) instead formulates dense warping estimation in a probabilistic framework, where prediction certainty is reflected by the variance of the predicted distribution.

## 2.3 EVIDENTIAL DEEP LEARNING IN COMPUTER VISION

Since its inception by Sensoy et al. (2018), evidential deep learning has been investigated on a wide range of computer vision tasks (Gao et al., 2024). As it is originally formulated for classification problem, its application to image classification task is straightforward. It has been shown to provide improved uncertainty estimation and detect out-of-distribution samples better than traditional neural networks (Sensoy et al., 2020; Hu et al., 2021). For object detection task, evidential uncertainty has been used for mining pseudo-unknown objects (Su et al., 2023). It has also been explored in segmentation task for uncertainty-guided rectification (Shi et al., 2024).

## 2.4 ROBUSTNESS ENHANCEMENTS

Most works on enhancing robustness of deep learning models have focused on the robustness to adversarial examples (Goodfellow et al., 2014). Adversarial training (Madry, 2017; Shafahi et al., 2019), in which adversarial examples are injected into training data, has been shown to withstand strong attacks. Instead of focusing on adversarial robustness, Hendrycks & Dietterich (2019) proposed benchmarks to evaluate a classifier's robustness to 15 types of common corruptions, and proposed to enhance corruption robustness by histogram equalization and more sophisticated network architectures. Michaelis et al. (2019) extended (Hendrycks & Dietterich, 2019) for benchmarking robustness in object detection and showed that stylizing training data leads to improved robustness.

## 3 METHOD

The system diagram of the proposed method is presented in Fig. 2. SotA dense matchers (Edstedt et al., 2023; 2024) typically consist of a global matcher that predicts a warp at coarse scale, and a series of warp refiners that refine the warp iteratively at finer scales. A certainty map is predicted along with the warp to indicate reliable matches. Previously, the certainty map is learnt by minimizing the binary cross-entropy (BCE) loss. In this work, we propose a deep evidential learning framework to learn the certainty map. In the following, we will first briefly describe the existing loss functions for training dense matchers, followed by details on how we incorporate evidential learning into dense matchers to train a more robust model.

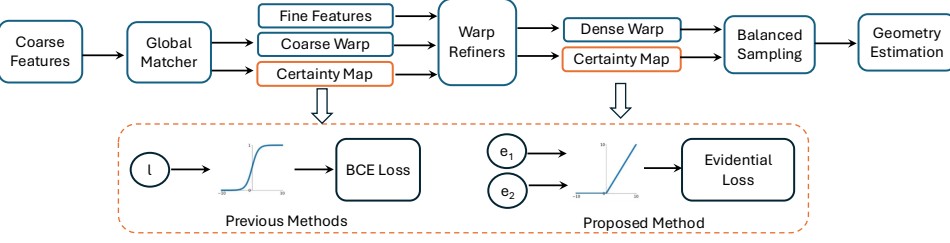

Figure 2: System diagram of the proposed method.

## 3.1 RECAP OF DENSE MATCHERS

Given two images $I^{\mathcal{A}}$ and $I^{\mathcal{B}}$, dense matchers predict a dense warp by first performing global matching on coarse-scale features and then refining the estimated warp on fine-scale features. Let $X^{\mathcal{A}}$ denote the pixels in $I^{\mathcal{A}}$, $X_{\mathcal{P}}^{\mathcal{A}} \in X^{\mathcal{A}}$ the subset of pixels having reliable matches in $I^{\mathcal{B}}$, and $x^{gt}$ indicates the corresponding coordinates in $I^{\mathcal{B}}$ for $x \in X_{\mathcal{P}}^{\mathcal{A}}$ . During training, the model is learned to minimize the following loss:

$$\mathcal{L} = \mathcal{L}_{\text{coarse}} + \mathcal{L}_{\text{fine}}. \tag{1}$$

The coarse-scale prediction performs global matching, which predicts the correspondence for each pixel in $I^{\mathcal{A}}$ and a certainty score for each correspondence. The coarse-scale loss is computed as:

$$\mathcal{L}_{\text{coarse}} = \frac{1}{|X_{\mathcal{P}}^{\mathcal{A}}|} \sum_{x \in X_{\mathcal{P}}^{\mathcal{A}}} \mathcal{L}_{gm}(gm(x), x^{gt}) + \lambda \frac{1}{|X^{\mathcal{A}}|} \sum_{x \in X^{\mathcal{A}}} \mathcal{L}_{BCE}(p_{\text{conf}}(x)), \tag{2}$$

where $gm(x)$ is the predicted correspondence at coarse scale, $\mathcal{L}_{gm}$ is global matching loss, which can be $\ell_2$ loss (Edstedt et al., 2023) or cross-entropy loss (Edstedt et al., 2024) depending on the formulation, $\mathcal{L}_{BCE}$ is the binary cross entropy loss, $p_{\text{conf}}$ is the certainty score obtained by applying sigmoid function on logits, and $\lambda = 0.01$ is a balanced term.

The fine-scale prediction estimates a residual offset for the warp and certainty score. The residuals are then added to the original estimation and the fine-scale loss is computed on the refined estimation:

$$\mathcal{L}_{\text{fine}} = \frac{1}{|X_{\mathcal{P}}^{\mathcal{A}}|} \sum_{x \in X_{\mathcal{P}}^{\mathcal{A}}} \mathcal{L}_{reg}(rw(x), x^{gt}) + \lambda \frac{1}{|X^{\mathcal{A}}|} \sum_{x \in X^{\mathcal{A}}} \mathcal{L}_{BCE}(p_{\text{conf}}(x)), \tag{3}$$

where $rw(x)$ is the refined warping coordinates, and $\mathcal{L}_{reg}$ is the regression loss between the predicted warp and ground truth warp. The fine-scale refinement is typically performed iteratively on multiple feature scales, and $\mathcal{L}_{\text{fine}}$ is computed for each scale and summed up to form the final loss.

## 3.2 CERTAINTY ESTIMATION BY EVIDENTIAL DEEP LEARNING

The warp predicted by dense matching methods (Edstedt et al., 2024; 2023) may contain inaccurate correspondences. It is necessary to estimate the certainty of each predicted correspondence and use those confident ones for downstream geometric matching tasks. In previous methods (Edstedt et al., 2024; 2023), certainty estimation is modeled as a binary classification problem, where pixels having reliable matches in $I^{\mathcal{B}}$ (*i.e.*, pixels whose projected depth to $I^{\mathcal{B}}$ are consistent with the ground truth depth) are assigned label 1, and pixels not having reliable matches in $I^{\mathcal{B}}$ are assigned label 0. The binary classifier is implemented as neural networks and trained by minimizing the cross entropy loss $\mathcal{L}_{BCE}$ as defined in Eq. (2) and Eq. (3).

In this work, we instead employ evidential deep learning for certainty estimation. Evidential learning considers a classification task as forming a subjective opinion (Josang, 2016) over the class label space. A subjective opinion can be formalized as an ordered triplet $\boldsymbol{\tau} = (\boldsymbol{b}, u, \boldsymbol{r})$, where $\boldsymbol{b}$ represents a belief mass distribution over the $K$ categories, $u$ the uncertainty mass and $\boldsymbol{r}$ a base rate (prior distribution) (Gao et al., 2024). The subjective logic theory establishes a bijection between a subjective opinion and a Dirichlet distribution $D(\boldsymbol{p}|\boldsymbol{\alpha})$, where the concentration parameters $\boldsymbol{\alpha}$ can be obtained from the triplet $\boldsymbol{\tau}$ as:

$$\boldsymbol{\alpha} = \boldsymbol{b}W_0/u + \boldsymbol{r}W_0, \tag{4}$$

where $W_0$ is a positive prior weight.

EDL employs deep neural networks to generate the subjective opinion. More specifically, the network is trained to generate a non-negative evidence vector $\boldsymbol{e} \in \mathbb{R}_+^K$, with each element $\boldsymbol{e}_i$ representing the amount of evidence supporting the claim that "the input sample belongs to the $i$-th category". The belief mass and uncertainty mass in the triplet can be computed from the evidence vector as:

$$\boldsymbol{b}_i = \frac{\boldsymbol{e}_i}{\sum_{j \in [K]} \boldsymbol{e}_j + W_0}, \quad u = \frac{W_0}{\sum_{j \in [K]} \boldsymbol{e}_j + W_0}. \tag{5}$$

Substituting $\boldsymbol{b}_i$ and $u$ into Eq. (4) gives $\boldsymbol{\alpha} = \boldsymbol{e} + \boldsymbol{r}W_0$. Sensoy et al. (2018) set the base rate $\boldsymbol{r}$ to be a uniform distribution, *i.e.*, $\boldsymbol{r}_i = 1/K$, and $W_0$ to be the class number $K$, which further simplifies

Eq. (4) and Eq. (5) as:

$$\boldsymbol{\alpha} = \boldsymbol{e} + 1, \quad \boldsymbol{b}_i = \frac{\boldsymbol{e}_i}{\sum_{j \in [K]} \boldsymbol{\alpha}_j}, \quad u = \frac{K}{\sum_{j \in [K]} \boldsymbol{\alpha}_j}. \tag{6}$$

By training the model to estimate the parameters of the evidential distribution, EDL represents the prediction of the neural network as a probability density function (PDF), rather than the point estimate of probability. The concentration parameters $\boldsymbol{\alpha}$ determines the spread of the PDF: the larger the value of $\boldsymbol{\alpha}$, the more concentrated the Dirichlet distribution, and the smaller the uncertainty (Eq. (6)). By modeling second-order probabilities and uncertainty, EDL is able to mitigate the over-confidence issue of neural network models and provide improved uncertainty estimation (Sensoy et al., 2018).

We modify the certainty prediction branch of dense matcher to output a two-dimensional evidence vector $\boldsymbol{e}$ for each pixel $x \in X^{\mathcal{A}}$, with $\boldsymbol{e}_1$ representing the amount of evidence supporting the claim that "the predicted correspondence is reliable" and $\boldsymbol{e}_2$ the amount of evidence supporting the claim vice-versa. The Dirichlet parameters can then be obtained by Eq. (6). We replace the cross entropy loss $\mathcal{L}_{BCE}$ used in Eq. (2) and Eq. (3) with the squared loss integrated over the Dirichlet distribution and a Kullback-Leibler (KL) divergence term for regularization:

$$\mathcal{L}_{el}(x) = \mathbb{E}_{\boldsymbol{p} \sim D(\boldsymbol{p}|\boldsymbol{\alpha}(x))} ||\boldsymbol{y}(x) - \boldsymbol{p}||^2 + KL[D(\boldsymbol{p}|\tilde{\boldsymbol{\alpha}}(x))||D(\boldsymbol{p}| < 1, \dots, 1 >)], \tag{7}$$

where $\boldsymbol{y}$ is the one-hot vector indicating the ground truth class, $D(\boldsymbol{p}|\boldsymbol{\alpha})$ denotes the Dirichlet distribution of class probabilities parameterized by $\boldsymbol{\alpha} = \boldsymbol{e} + 1$, and $\tilde{\boldsymbol{\alpha}} = \boldsymbol{y} + (1 - \boldsymbol{y}) \odot \boldsymbol{\alpha}$ is the Dirichlet parameters after removing the evidence of ground truth class. The KL divergence term aims to shrink the evidence of non-ground truth class to zero.

With the predicted Dirichlet distribution of class probabilities, the expected probability for each class is obtained by computing the mean over the Dirichlet distribution (Sensoy et al., 2018):

$$\boldsymbol{p}(x) = \frac{\boldsymbol{\alpha}(x)}{S}, \tag{8}$$

where $S = \sum_{i=1}^{K} \boldsymbol{\alpha}_i(x)$ and $K = 2$. The expected probability for class 1 is then used as certainty score to indicate the reliability of each predicted correspondence.

## 4 EXPERIMENTS

### 4.1 IMPLEMENTATION DETAILS

**Model Architecture** We use the SotA dense matcher RoMa (Edstedt et al., 2024) as our baseline. We modify the certainty prediction branch of RoMa to output a 2-dimensional evidence vector for each pixel, and replace $\mathcal{L}_{BCE}$ used in Eq. (2) and Eq. (3) with the evidential loss defined in Eq. (7) to train the network to accumulate evidence.

**Training Setup** We follow the training setup of RoMa (Edstedt et al., 2024), where the batch size is 32, encoder learning rate is $10^{-4}$, decoder learning rate is $5 \cdot 10^{-6}$, and the model is trained for 250,000 iterations. The training image resolution is $560 \times 560$. For outdoor geometry estimation, the model is trained on the MegaDepth dataset (Li & Snavely, 2018) using the same training and test splits as in RoMa. For indoor geometry estimation, the model is trained on a combination of MegaDepth and ScanNet datasets (Dai et al., 2017a) in a similar fashion as previous works (Edstedt et al., 2024; Sun et al., 2021).

**Test Setup** We use the same test setting as RoMa to ensure fair comparison. For balanced sampling, we use score threshold 0.05 and sample 5,000 matches, the same as DKM (Edstedt et al., 2023) and RoMa. When benchmarking with other methods, we use the released outdoor/indoor model for outdoor/indoor geometry estimation and follow the default settings.

### 4.2 BENCHMARK DATASETS

MegaDepth-1500 (Sun et al., 2021) is a popular outdoor benchmark consisting of 1500 image pairs from scene 0015 and 0022. To evaluate each method's robustness under image corruptions, we

create the MegaDepth-1500-C benchmark, where the image pairs in MegaDepth-1500 are distorted by the 15 corruption types proposed in (Hendrycks & Dietterich, 2019) at five severity levels. The 15 corruption types are sorted into four categories: noise, blur, weather and digital, and are meant to measure a model's robustness against corruptions unseen during training. For indoor benchmark, we adopt the popular ScanNet-1500 (Sarlin et al., 2020) and create the corrupted version ScanNet-1500-C in a similar way as MegaDepth-1500-C.

## 4.3 GEOMETRY ESTIMATION ON CORRUPTED DATA

**MegaDepth-1500-C Pose Estimation** We follow the protocol in (Edstedt et al., 2024; Sun et al., 2021) to estimate camera pose and report AUC at different thresholds. Fig. 3 presents the AUC@5° for each corruption type at each severity level. We observe dense matching methods DKM and RoMa significantly outperform the detector-free method LoFTR over all corruptions and severity levels. Our method further improves upon RoMa. For the category of noise corruption (Gaussian noise, shot noise and impulse noise), we observe larger performance gap at higher severity level, achieving 10.1% improvement for the shot noise corruption at severity level 5. The results averaged over severity levels and corruption types are reported in Table 1. Our method achieves 2.2% improvement in mean corruption AUC (mCA) compared to RoMa. Note that on clean data our method achieves only 0.4% gain over RoMa. The increase in mCA thus indicates improved corruption robustness for our method.

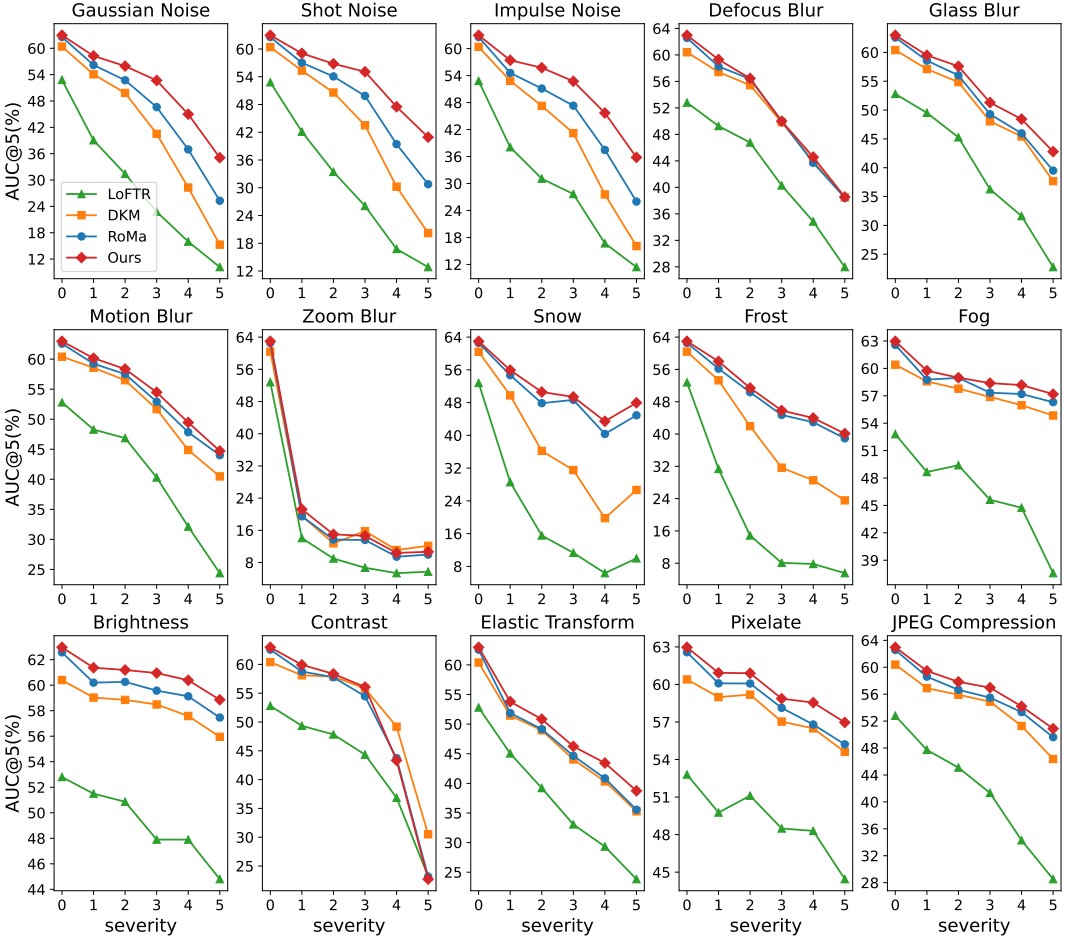

Figure 3: AUC@5°(%) for each corruption type in MegaDepth-1500-C.

**MegaDepth-1500 under Adversarial Attacks** We further consider adversarial attacks, which is an extreme type of corruption that perturbs the image in carefully crafted direction and serves as worst-case analysis for model robustness. Two attacks, Fast Gradient Sign Method (FGSM) (Goodfellow

Table 1: AUC@5°(%) averaged over severity levels and corruption types in MegaDepth-1500-C. The value for each corruption type is the value averaged over severity levels, and mCA is the mean corruption AUC@5° for all 15 corruption types. Models are trained only on clean images.

| Method | Clean | mCA | Noise | | | Blur | | | | Weather | | | | Digital | | | |
|---|---|---|---|---|---|---|---|---|---|---|---|---|---|---|---|---|---|
| | | | Gauss. | Shot | Impulse | Defocus | Glass | Motion | Zoom | Snow | Frost | Fog | Bright | Contrast | Elastic | Pixel | JPEG |
| LoFTR | 52.8 | 32.2 | 23.9 | 26.2 | 25.0 | 39.8 | 37.1 | 38.4 | 8.2 | 14.3 | 13.5 | 45.2 | 48.6 | 40.3 | 34.1 | 48.4 | 39.4 |
| DKM | 60.4 | 44.3 | 37.6 | 40.0 | 37.0 | 49.0 | 48.6 | 50.4 | 14.3 | 32.8 | 35.8 | 56.8 | 58.0 | **50.3** | 44.0 | 57.3 | 53.1 |
| RoMa | 62.6 | 47.6 | 43.6 | 46.2 | 43.3 | 49.4 | 49.9 | 52.3 | 13.2 | 47.3 | 46.6 | 57.7 | 59.3 | 47.6 | 44.4 | 58.1 | 54.7 |
| Ours | **63.0** | **49.8** | **49.4** | **51.9** | **49.5** | **49.8** | **51.9** | **53.4** | **14.4** | **49.5** | **47.9** | **58.5** | **60.5** | 48.1 | **46.6** | **59.2** | **55.9** |

et al., 2014) and Projected Gradient Descent (PGD) (Madry, 2017), are investigated. The attacks are $\ell_\infty$-bounded and the PGD attack is computed with 20 iterations. We experiment with different perturbation budgets (denoted as $\epsilon$) and the results are presented in Fig. 4. We observe that the detector-free approach LoFTR is vulnerable to adversarial attacks, with AUC quickly dropping to zero as $\epsilon$ increases. Compared to DKM, RoMa has better adversarial robustness under PGD attack, probably due to the use of more sophisticated network architectures. Our method is able to consistently outperform others, achieving up to 2.9% gain over RoMa for the FGSM attack, and up to 3.4% gain over RoMa for the PGD attack.

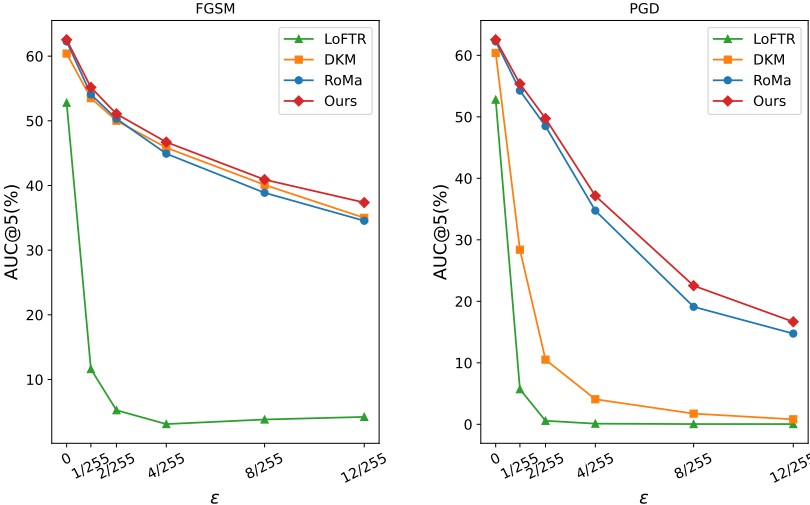

Figure 4: AUC@5°(%) on MegaDepth-1500 under FGSM and PGD attacks with different $\epsilon$ values (perturbation budgets).

**ScanNet-1500-C and ScanNet-1500 under Adversarial Attacks** The benchmarking results on ScanNet-1500-C (with the same corruption settings as MegaDepth-1500-C) and ScanNet-1500 under adversarial attacks (using the same attack settings as MegaDepth-1500) are presented in Table 2 and Fig. 5, respectively. While the advantage of our method over RoMa is less pronounced (refer to Section 4.5 for the analysis of the underlying reasons), we still achieve up to 0.6% gain over RoMa on ScanNet-1500-C, and up to 0.7% gain for the FGSM attack.

Table 2: AUC@5°(%) averaged over severity levels and corruption types in ScanNet-1500-C. The value for each corruption type is the value averaged over severity levels, and mCA is the mean corruption AUC@5° for for all 15 corruption types. Models are trained only on clean images.

| Method | Clean | mCA | Noise | | | Blur | | | | Weather | | | | Digital | | | |
|---|---|---|---|---|---|---|---|---|---|---|---|---|---|---|---|---|---|
| | | | Gauss. | Shot | Impulse | Defocus | Glass | Motion | Zoom | Snow | Frost | Fog | Bright | Contrast | Elastic | Pixel | JPEG |
| LoFTR | 22.1 | 12.3 | 13.2 | 14.3 | 13.1 | 15.3 | 18.4 | 16.0 | 1.8 | 4.4 | 2.6 | 5.3 | 16.4 | 6.1 | 19.2 | 21.2 | 17.7 |
| DKM | 29.4 | 22.3 | 21.8 | 22.5 | 21.1 | 28.6 | 28.4 | 28.0 | 7.9 | 14.1 | 9.3 | 22.5 | 26.1 | 22.5 | 27.3 | 29.5 | 25.2 |
| RoMa | 31.8 | 25.9 | 25.8 | 26.3 | 25.4 | 30.5 | 30.4 | 30.1 | 9.9 | 22.0 | 15.4 | 27.6 | 29.3 | 26.1 | 29.8 | 31.2 | 28.4 |
| Ours | **32.2** | **26.3** | **26.2** | **26.6** | **26.1** | **30.7** | **30.6** | **30.5** | **10.2** | **22.6** | **15.5** | **28.1** | **29.3** | **26.7** | **30.0** | **31.8** | **28.8** |

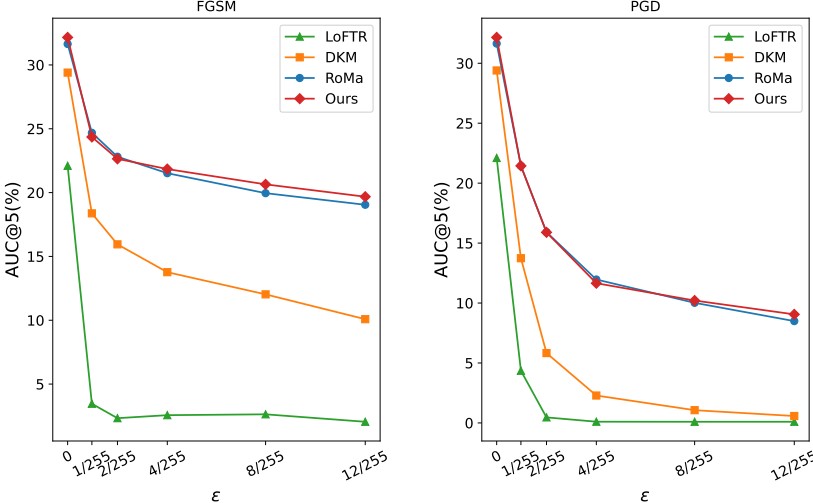

Figure 5: AUC@5°(%) on ScanNet-1500 under FGSM and PGD attacks with different $\epsilon$ values (perturbation budgets).

## 4.4 GEOMETRY ESTIMATION ON CLEAN DATA

The benchmarking results of our method on the original clean MegaDepth-1500 and ScanNet-1500 benchmarks are presented in Table 3 and Table 4. Our method achieves comparable, or slightly better performance compared to RoMa, suggesting that the improved robustness of our method does not sacrifice the performance on clean data.

Table 3: SotA comparison on MegaDepth-1500. Measured in AUC (higher is better).

| Method ↓          AUC@ → | 5° ↑ | 10° ↑ | 20° ↑ |
|---|---|---|---|
| LightGlue (Lindenberger et al., 2023) ICCV'23 | 51.0 | 68.1 | 80.7 |
| LoFTR (Sun et al., 2021) CVPR'21 | 52.8 | 69.2 | 81.2 |
| PDC-Net+ (Truong et al., 2023) TPAMI'23 | 51.5 | 67.2 | 78.5 |
| ASpanFormer (Chen et al., 2022) ECCV'22 | 55.3 | 71.5 | 83.1 |
| ASTR (Yu et al., 2023) CVPR'23 | 58.4 | 73.1 | 83.8 |
| DKM (Edstedt et al., 2023) CVPR'23 | 60.4 | 74.9 | 85.1 |
| PMatch (Zhu & Liu, 2023) CVPR'23 | 61.4 | 75.7 | 85.7 |
| CasMTR (Cao & Fu, 2023) ICCV'23 | 59.1 | 74.3 | 84.8 |
| RoMa (Edstedt et al., 2024) CVPR'24 | 62.6 | 76.7 | 86.3 |
| Ours | **63.0** | **76.9** | **86.5** |

Table 4: SotA comparison on ScanNet-1500. Measured in AUC (higher is better).

| Method ↓          AUC@ → | 5° ↑ | 10° ↑ | 20° ↑ |
|---|---|---|---|
| SuperGlue (Sarlin et al., 2020) CVPR'19 | 16.2 | 33.8 | 51.8 |
| LoFTR (Sun et al., 2021) CVPR'21 | 22.1 | 40.8 | 57.6 |
| PDC-Net+ (Truong et al., 2023) TPAMI'23 | 20.3 | 39.4 | 57.1 |
| ASpanFormer (Chen et al., 2022) ECCV'22 | 25.6 | 46.0 | 63.3 |
| PATS (Ni et al., 2023) CVPR'23 | 26.0 | 46.9 | 64.3 |
| DKM (Edstedt et al., 2023) CVPR'23 | 29.4 | 50.7 | 68.3 |
| PMatch (Zhu & Liu, 2023) CVPR'23 | 29.4 | 50.1 | 67.4 |
| CasMTR (Cao & Fu, 2023) ICCV'23 | 27.1 | 47.0 | 64.4 |
| RoMa (Edstedt et al., 2024) CVPR'24 | 31.8 | 53.4 | 70.9 |
| Ours | **32.2** | **54.1** | **71.6** |

## 4.5 FURTHER ANALYSIS

**Why Our Method Outperforms RoMa** We conduct further analysis on our method to understand why it can outperform RoMa. We compute the average endpoint error (AEPE) (Melekhov et al., 2019), which is defined as the average Euclidean distance between the estimated and ground truth warp. The AEPE for our method and RoMa on MegaDepth-1500-C is 6.12 and 6.07, respectively. The comparable AEPE values suggest that the warp prediction branch of our method performs similarly to RoMa. This is understandable as we only modify the certainty prediction branch of RoMa for evidential learning. We visualize the certainty map predicted by RoMa and our method in Fig. 6. Compared to clean images, corruption causes both RoMa and our method to produce reduced certainty in matchable regions. However, RoMa tends to predict very low certainty value (*e.g.*, below 0.05) for matchable regions on corrupted images, *i.e.*, the model over-confidently classifies the pixels as unmatchable. As the following balanced sampling algorithm samples matches based on the estimated certainty score, the extremely low certainty values make it hard to sample enough matches from matchable regions. Our method is able to predict higher certainty value (*i.e.*, $\sim 0.4$) for matchable region. Such prediction, though not perfect, can still facilitate the following balanced sampling step to sample a diverse set of matches from matchable regions. This problem is less severe on ScanNet, where the overlap between the image pairs is larger and the sampled matches can still well cover the matchable region even though the predicted certainty is low. This may explain why the advantage of our method is less significant on corrupted ScanNet-1500 compared to corrupted MegaDepth-1500.

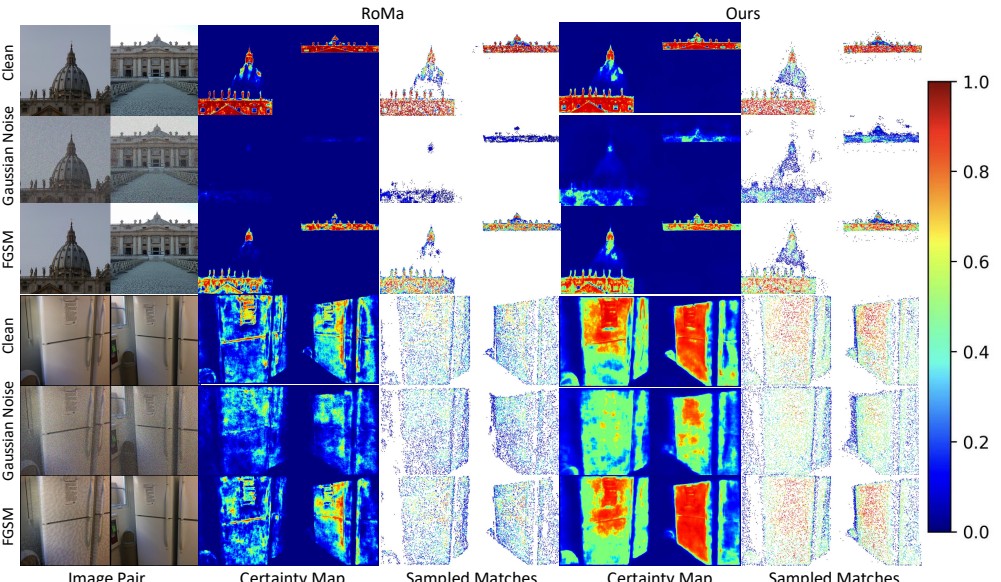

Figure 6: Visualization of certainty map and sampled matches for pose estimation. Top scene is from MegaDepth-1500 and bottom scene is from ScanNet-1500. Our method predicts higher certainty value for matchable region on corrupted images than RoMa, enabling the following balanced sampling step to select a set of diverse and reliable matches for downstream geometry estimation tasks.

To cast more insight into the behavior of our method, we visualize the evidence map output by our model in Fig. 7. We observe that under corrupted cases, the model cannot predict high evidence for both classes in the matchable region, resulting in similar probability for the two classes, *i.e.*, there is high uncertainty in the estimation. Compared to RoMa's over-confidence estimation, a high-uncertainty prediction still assigns some confidence to the correct class and causes less devastating effect to the following geometry estimation pipeline.

**Ablation Study on Applying EDL at Different Scales** We report the ablation study on applying EDL at different scales in Table 5. We observe that applying EDL on coarse scale alone is not effective, as the final prediction comes from the finest scale (which is still learnt by the BCE loss).

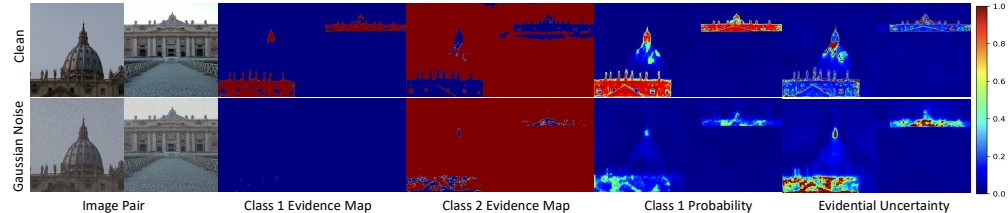

Figure 7: Visualization of evidence map predicted by our model. Class 1 represents the claim "the predicted correspondence is reliable" and Class 2 represents the opposite. Class 1 probability is essentially the certainty score. The last column shows the evidential uncertainty (the built-in uncertainty defined in Eq. (6)), which indicates high-uncertainty prediction in matchable regions on corrupted images. Note that the evidence value is unbounded non-negative value and has been clamped to [0,1] for visualization purposes.

Applying EDL on fine scales improves the performance on corrupted samples significantly. The best performance is achieved when EDL is applied on both coarse and fine scales.

Table 5: AUC@5°(%) on MegaDepth-1500 for models trained with EDL at different scales. The first row with both coarse and fine scale EDL disabled degenerates to the original RoMa model. Note that we use a reduced training setting for this study (training image resolution $448 \times 448$, batch size 16 and training iteration 62,500), so the reported performance is lower than those reported in Table 1.

| Coarse | Fine | AUC@5° (clean) | AUC@5° (Gaussian noise@5) |
|:------:|:----:|:--------------:|:--------------------------:|
|        |      | 60.9           | 22.8                       |
| ✓      |      | 59.9           | 22.9                       |
|        | ✓    | 61.4           | 31.7                       |
| ✓      | ✓    | **61.5**       | **33.1**                   |

**Computation Cost Comparison** We report the training and inference time of RoMa and our method in Table 6. For training, our method takes 1.4% more GPU hours than RoMa. For inference, our method actually incurs marginally lower cost. This is probably due to the fact that EDL uses simple operations (*e.g.*, addition and division) to obtain the final probability, eliminating the more complicated sigmoid computation in RoMa.

Table 6: Comparison of computation time. Training is conducted on 8 A40 GPUs with batch size 32, input size $560 \times 560$ and 250k iterations. Inference is conducted on 1 A40 GPU with batch size 1 (1 pair of images) and input size $672 \times 672$. The inference time is averaged over the 1500 pairs in MegaDepth-1500 benchmark.

|      | Training (in hours) | Inference (in ms) |
|------|:-------------------:|:-----------------:|
| RoMa | 126.6               | 329               |
| Ours | 128.4               | 327               |

## 5 CONCLUSIONS

In this work, we propose to incorporate evidential deep learning into the certainty prediction branch of dense feature matching methods. To evaluate the robustness of feature matching methods, we propose MegaDepth-1500-C and ScanNet-1500-C benchmarks that contain images distorted by 15 corruption types at five severity levels. Evaluation results show that EDL allows the model to predict higher certainty values for matchable region on corrupted images compared to traditional cross-entropy loss, facilitating the sampling of a set of both certain and diverse matches for geometry estimation. Our method achieved up to 10.1% improvement for noise corruption in MegaDepth-1500-C. We further investigate our method's robustness against adversarial attacks, and show that our method achieved up to 2.9% gain over SotA for the FGSM attack, and up to 3.4% gain for the PGD attack. These results demonstrate the effectiveness of EDL in improving the robustness of dense feature matching models.

ACKNOWLEDGMENTS

The authors would like to thank the reviewers for their constructive feedback during the review process. This work is supported by the Agency for Science, Technology and Research (A*STAR) under its MTC Programmatic Funds (Grant No. M23L7b0021).

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

## A APPENDIX

### A.1 IMPLEMENTATION DETAILS

**Model Architecture** The RoMa model consists of three modules: feature extractor, global matcher and warp refiners. The feature extractor uses frozen foundation model DINOv2 (Oquab et al., 2023) for extracting feature maps at stride 16, and VGG19 for extracting feature maps at stride [8, 4, 2, 1]. The global matcher consists of a Gaussian Process-based encoder and a transformer-based decoder. It takes as input feature maps at stride 16, and outputs initial warp and certainty map estimation. The warp refiner is CNN module that consists of $5 \times 5$ depth-wise convolution and $1 \times 1$ convolution. It is applied to features maps at scale [16, 8, 4, 2, 1] iteratively. At each scale, it takes the corresponding feature maps and previously estimated warp and certainty map as input and outputs a residual estimation on warp and certainty. To modify RoMa for evidential learning, we increase the number of the output channel in the transformer-based decoder and the warp refiner by 1. The original channel for sigmoid logit and the added channel are then used for the 2-class evidence estimation in our proposed formulation.

**Details on Datasets** The MegaDepth dataset (Li & Snavely, 2018) contains images collected from Internet for 196 outdoor scenes. For each scene, the camera pose and depth map of each image are computed by COLMAP, a state-of-the-art software for Structure-from-Motion (SfM) and Multi-View Stereo (MVS). We use the same training split as in RoMa. The ScanNet dataset (Dai et al., 2017a) contains 1513 RGB-D scans for indoor scenes. The camera pose of each image is estimated by the BundleFusion system (Dai et al., 2017b). The training and test split of the dataset contains 1201 and 312 scenes, respectively.

**Training Procedure** We follow the training setup in RoMa's repo (https://github.com/Parskatt/RoMa). The batch size is 32 and training image size is $560 \times 560$. The learning rate for encoder (including the VGG19 feature extractor. The DINOv2 weights are frozen) is $10^{-4}$, and the learning rate for decoder (including the global matcher and warp refiners) is $5 \cdot 10^{-6}$. The model is trained for 250k iterations using the AdamW (Loshchilov, 2017) optimizer with weight decay 0.01, and the learning late is decayed by 0.1 at 225k iteration. For outdoor geometry estimation, training pairs are randomly sampled from the MegaDepth training split. For indoor geometry estimation, training pairs are alternatively sampled from the MegaDepth or the ScanNet training split, following (Edstedt et al., 2024; Sun et al., 2021). Training takes approximately 5 days on a server with 8 A40 GPUs.

A.2 ROBUSTNESS AGAINST NEWER CORRUPTIONS AND ATTACKS

**3D Common Corruptions (3DCC)** We evaluate our method on three corruptions from the 3DCC collections (Kar et al., 2022), including low light noise, ISO noise and color quantization. Our method achieves consistent and significant gains over RoMa, obtaining up to 9.4% increase in AUC@5°. Some examples of corrupted images are shown in Fig. 9. We notice that for those corruptions that require depth information for generation, *e.g.*, 3D motion blur, Fog 3D, using MegaDepth's ground truth depth map does not create realistic images, and thus are not used in our benchmarking. This suggests that 3D corrupted image generation requires dataset-specific parameter tuning, lacking the plug-and-play convenience as its 2D counterpart.

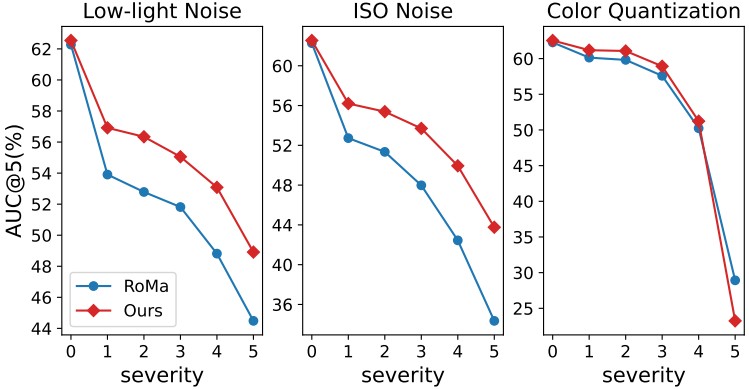

Figure 8: AUC@5°(%) for three corruption types in MegaDepth-1500-3DCC.

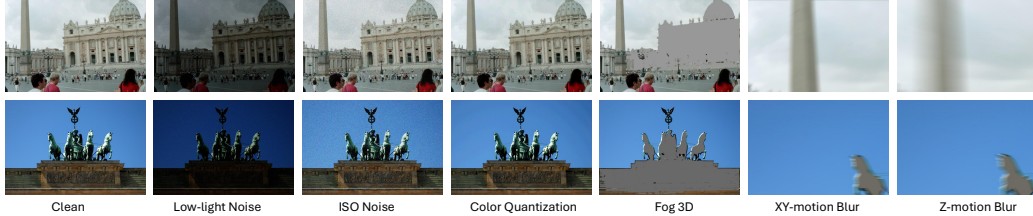

Figure 9: Corrupted images from MegaDepth-1500-3DCC. For corruptions that require depth information for generation, *e.g.*, 3D motion blur, Fog 3D, 3DCC cannot work out-of-the-shelf.

**Pixel-wise Adversarial Attacks** We experiment with a recently proposed attack CosPGD (Agnihotri et al., 2024). CosPGD is an improved version of PGD, where the pixel-wise loss is weighted by the cosine similarity between (normalized) prediction and ground truth so that pixels with correct prediction will be perturbed more in generating the attack. CosPGD can be readily applied for both classification and regression tasks. For regression task, the prediction and ground truth are normalized by softmax before computing the cosine similarity. We evaluate the robustness of RoMa and our method under CosPGD attack and the results are shown in Fig. 10. Our method outperforms RoMa consistently, achieving up to 3.0% gain over RoMa.

A.3 BENCHMARKING ON ADDITIONAL DATASETS

We provide additional benchmarking results in Table 7 and Table 8. Our method obtains 1.7% increase in mAA@10px than RoMa on WxBS (Mishkin et al., 2015), and 0.5% increase in AUC@3px on HPatches (Balntas et al., 2017).

A.4 VERIFY OUR METHOD WITH OTHER DENSE MATCHING MODELS

In the main experiments, we use the SotA dense matcher RoMa as baseline and apply our proposed framework on it. Here we apply our method on another dense matcher DKM (Edstedt

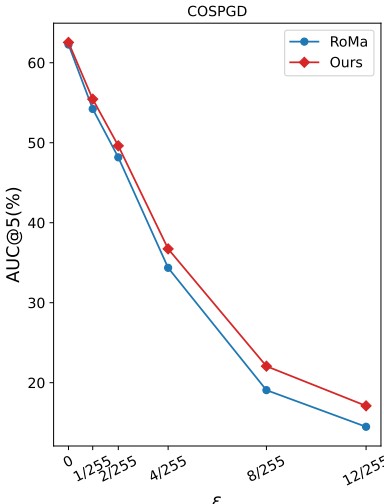

Figure 10: AUC@5°(%) on MegaDepth-1500 under CosPGD attack with different $\epsilon$ values (perturbation budgets).

Table 7: SotA comparison on WxBS (Mishkin et al., 2015). Measured in mAA@10px (higher is better).

| Method          mAA@ $\rightarrow$ | 10px $\uparrow$ |
|---|---|
| DISK (Tyszkiewicz et al., 2020) NeurIPS'20 | 35.5 |
| DISK + LightGlue (Tyszkiewicz et al., 2020; Lindenberger et al., 2023) ICCV'23 | 41.7 |
| SuperPoint +SuperGlue (DeTone et al., 2018; Sarlin et al., 2020) CVPR'20 | 31.4 |
| LoFTR (Sun et al., 2021) CVPR'21 | 55.4 |
| DKM (Edstedt et al., 2023) CVPR'23 | 58.9 |
| RoMa (Edstedt et al., 2024) CVPR'24 | 80.1 |
| **Ours** | **81.8** |

et al., 2023). Similar to RoMa, DKM employs a certainty estimation branch in both global matcher and warp refiners. We modify the certainty estimation branch to generate evidence and replace the BCE loss with evidential loss. We follow the training setup in DKM's repo (https://github.com/Parskatt/DKM). The model is trained on a server with 8 A5000 GPUs with batch size 16, image resolution $540 \times 720$ and training iteration 500k. Training takes roughly 5 days. For fair comparison, we retrain the original DKM model using the same setup and report the results in Table 9. Our method increases the clean AUC@5° by 1.6% on MegaDepth-1500, and increases the

Table 8: Homography estimation on HPatches (Balntas et al., 2017), measured in AUC (higher is better).

| Method $\downarrow$          AUC $\rightarrow$ | @3px | @5px | @10px |
|---|---|---|---|
| SuperGlue (Sarlin et al., 2020) CVPR'19 | 53.9 | 68.3 | 81.7 |
| LoFTR (Sun et al., 2021) CVPR'21 | 65.9 | 75.6 | 84.6 |
| TopicFM (Giang et al., 2023) Arxiv'22 | 67.3 | 77.0 | 85.7 |
| 3DG-STFM (Mao et al., 2022) ECCV'22 | 64.7 | 73.1 | 81.0 |
| ASpanFormer (Chen et al., 2022) ECCV'22 | 67.4 | 76.9 | 85.6 |
| PDC-Net+ (Truong et al., 2021) Arxiv'21 | 67.7 | 77.6 | 86.3 |
| DKM (Edstedt et al., 2023) CVPR'23 | 71.3 | 80.6 | 88.5 |
| RoMa (Edstedt et al., 2024) CVPR'24 | 72.7 | 81.4 | 89.1 |
| **Ours** | **73.2** | **81.9** | **89.3** |

mean corruption AUC@5° by 1.3% on MegaDepth-1500-C. These results demonstrate the generalizability of our method beyond RoMa.

Table 9: Applying our method with the DKM model. The reported performance is AUC@5°(%) averaged over severity levels and corruption types on MegaDepth-1500-C.

| Method | Clean | mCA | Noise | | | Blur | | | | Weather | | | | Digital | | | |
|---|---|---|---|---|---|---|---|---|---|---|---|---|---|---|---|---|---|
| | | | Gauss. | Shot | Impulse | Defocus | Glass | Motion | Zoom | Snow | Frost | Fog | Bright | Contrast | Elastic | Pixel | JPEG |
| DKM | 58.4 | 39.8 | 31.1 | 33.4 | 29.0 | 44.5 | 42.9 | 46.4 | 11.5 | 27.4 | 30.6 | 53.8 | 54.9 | **48.1** | 41.2 | 53.6 | 49.7 |
| Ours | **60.0** | **41.1** | **31.2** | **33.5** | **29.0** | **46.8** | **45.9** | **48.4** | **13.4** | **27.7** | **33.3** | **54.8** | **56.6** | 47.1 | **43.1** | **55.7** | **50.8** |

## A.5 COMBINING CERTAINTY SCORE OF OUR METHOD WITH WARP OF ROMA

We report in Table 10 the results of combining certainty score from our model with the warp from the original RoMa model. We observe that using our certainty score, the performance of RoMa is increased from 25.3% to 36.5%, close to our results of 37.9%. In Section 4.5, we report the average endpoint error (AEPE), which is defined as the average Euclidean distance between the estimated and ground truth warp. The AEPE for our method and RoMa on MegaDepth-1500-C is 6.12 and 6.07, respectively. The comparable AEPE values suggest that the warp prediction branch of our method performs similarly to RoMa. These two studies combined suggest that it is the better certainty estimation that brings the improvement in performance.

Table 10: AUC@5°(%) for various combinations of certainty score and warp estimation on MegaDepth-1500.

| Cert (RoMa) | Warp (RoMa) | Cert (Ours) | Warp (Ours) | AUC@5° (clean) | AUC@5° (Gaussian noise@5) |
|---|---|---|---|---|---|
| ✓ | ✓ | | | 62.6 | 25.3 |
| | ✓ | ✓ | | 62.4 | 36.5 |
| | | ✓ | ✓ | 63.0 | 37.9 |

## A.6 EFFECT OF SCORE THRESHOLD

We experiment with different score thresholds for balanced sampling and the results are presented in Fig. 11. We observe that inappropriate choice of the score threshold can degrade the performance for both RoMa and our method, yet our method is still able to outperform RoMa over a wide range of values.

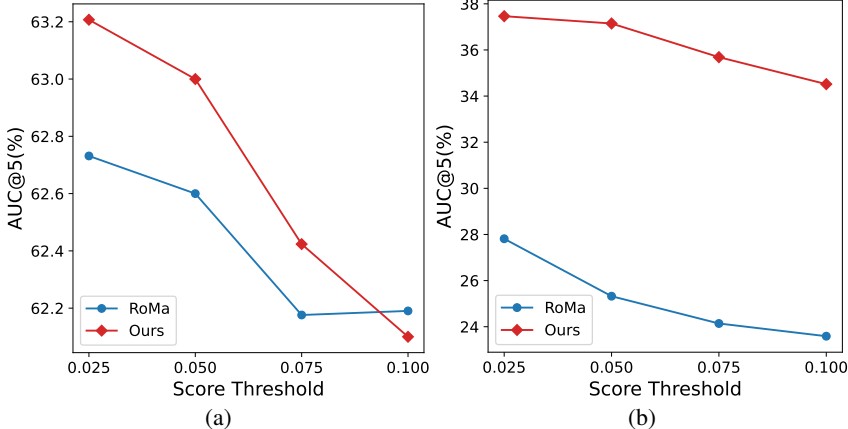

(a)  (b)

Figure 11: Effect of score threshold for balanced sampling. (a) AUC@5°(%) on MegaDepth-1500. (b) AUC@5°(%) on MegaDepth-1500 corrupted by Gaussian noise at severity level 5.

## A.7 QUALITATIVE RESULTS

More visualization results are presented in Fig. 12. Our method predicts higher certainty value for matchable region than RoMa across different corruption types.

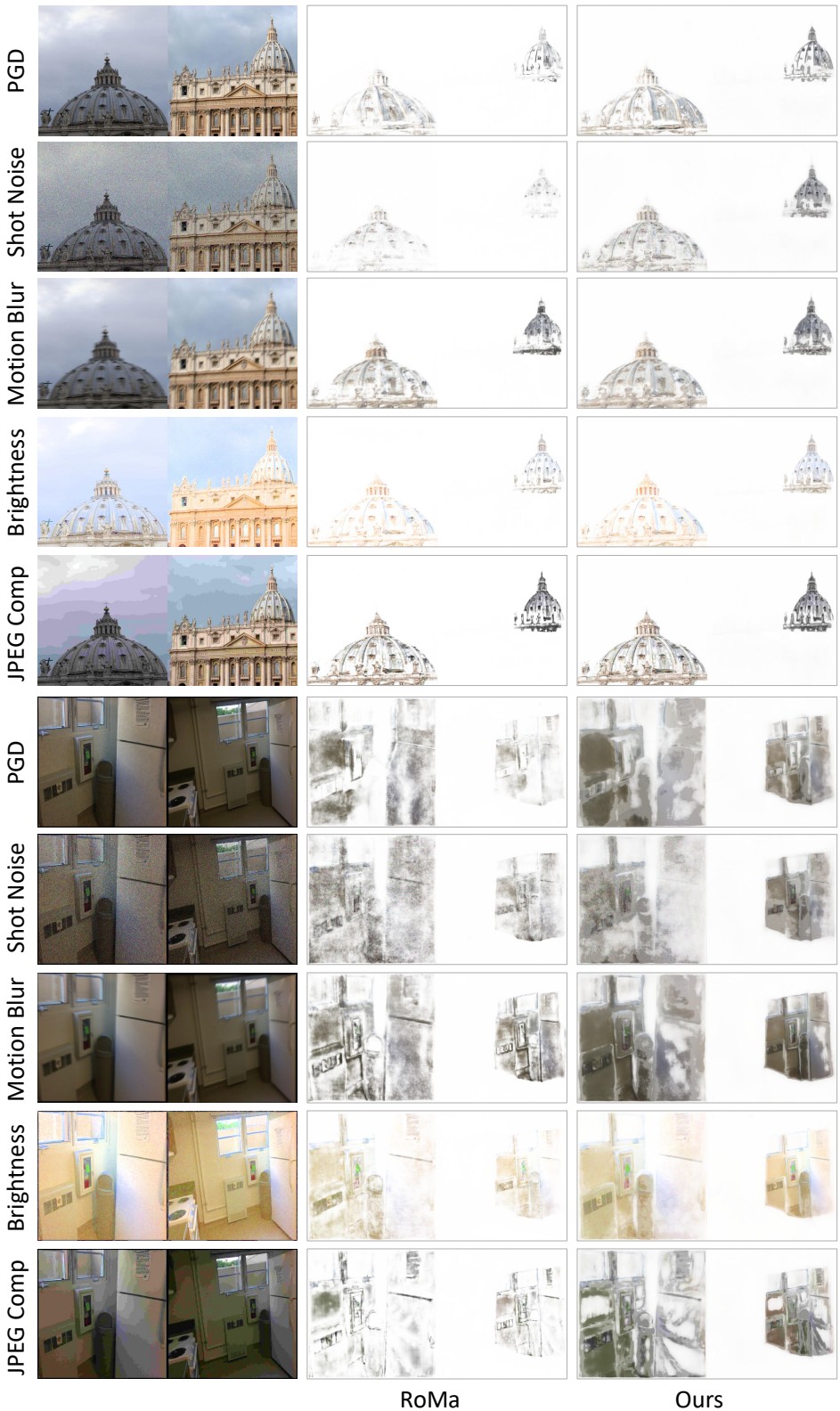

Figure 12: Qualitative results. The matching results are visualized by taking the pixel values from the other image using the estimated correspondences, and weighting by the estimated certainty. Top scene is from MegaDepth-1500 and bottom scene is from ScanNet-1500. Our method predicts higher certainty value for matchable region than RoMa across different corruption types.

