# OpenReview forum: "Evidential Learning-based Certainty Estimation for Robust Dense Feature Matching"
_ICLR.cc/2025/Conference — ICLR 2025 Poster_

### Official Review · Reviewer_Uwz6 · 2024-10-27

**Soundness:** 3
**Presentation:** 2
**Contribution:** 2
**Rating:** 5
**Confidence:** 4

**Summary:**

This paper proposes to unify evidential deep learning (EDL) and dense feature matching, achieving more robust matching results, especially for corrupted image pairs. The authors propose MegaDepth-1500-C and ScanNet-1500-C benchmarks to evaluate the robustness of the proposed method under common image corruptions. The proposed method enjoys superior results in both clean and corrupted data.

**Strengths:**

1. The incorporation of EDL to dense feature matching is interesting, and has not been investigated before.
2. The proposed method enjoys good performance in corrupted data.

**Weaknesses:**

1. Although the point of EDL is interesting, the usage of EDL for certainty estimation in dense feature matching is still questionable. From the introduction in Section 2.3, I think EDL's main advance is to detect out-of-distribution samples or mining pseudo-unknown objects. However, the certainty estimation in feature matching is just a binary classification task (matched or not matched). Why is EDL still effective? The authors did not provide a more insightful discussion about this key question.

2. The overall contribution is limited. Because of lacking enough in-depth discussion about EDL and certainty estimation in feature matching, makes this work appear more as a mere combination of these two approaches rather than a convincing exploration.

3. The introduction of EDL in Section 3.2 is insufficient, missing the necessary background/preliminary in related works.

4. Experiments are not sufficient, missing discussion of visual localization (InLoc and AachenDay-Night) and homography estimation (HPatches). The proposed method achieves significant improvements in corrupted data, while the improvements based on clean data are limited. As a general certainty estimation, the usage of EDL should consistently improve the matching accuracy in all scenarios.

**Questions:**

If EDL is mainly used for certainty estimation, what are the differences or relationships of the proposed method compared to outlier filtering post-processing in feature matching (RANSAC)?

---

> ### Author Response · Authors · 2024-11-28
>
> * W1. Why EDL is still effective for binary classification task
>
> As mentioned by the reviewer in the comments, EDL's main advantage is to detect out-of-distribution (OOD) samples or mining pseudo-unknown objects. This advantage comes from the fact that by modeling second-order probabilities and uncertainty, EDL provides improved uncertainty estimation. As shown in the seminal work [Sensoy et al.,  NeurIPS2018], EDL will produce larger uncertainty for OOD samples, while the standard approach (training the network by minimizing cross-entropy loss) tends to produce over-confident (low-uncertainty) predictions. Assuming a binary classification task and an OOD sample with ground truth label [1, 0], the standard approach may produce an over-confident prediction [0.01, 0.99], while EDL may produce a high-uncertainty prediction [0.4, 0.6]. Now assume that the system needs to continue the downstream estimation pipeline with the over-confident or high-uncertainty prediction, with which prediction shall the system perform better? Intuitively, an over-confident prediction is more devastating as the system cannot recover from it, while a high-uncertainty prediction still assigns some confidence to the correct class and the system may still be able to perform well with it. This is exactly what happens when we apply EDL for the certainty estimation task in dense matching. In Fig.6 of the paper, we provide visualization of the certainty map estimated by RoMa and our method. RoMa tends to predict very low certainty value (e.g., below 0.05) for matchable regions on corrupted images (i.e., over-confidently classify the region as unmatchable), while our model  produce a certainty value around 0.4 for matchable regions -- it is not perfect, but good enough to facilitate the following balanced sampling step to sample a diverse set of matches from matchable regions. We have modified the Introduction section and Section 4.5 to illustrate this point better.
>
>
> * W2. The overall contribution is limited, lacking enough in-depth discussion
>
> We added Fig.7 in the revised paper to cast more insight into the behavior of EDL in dense feature matching. We visualize the evidence map estimated by our model, and reveal that under corrupted cases, the model cannot produce high evidence for both classes in the matchable region. This cause high-uncertainty estimation for the matchable region. Compared to the over-confident prediction generated by RoMa, the EDL's prediction enables more effective sampling of reliable matches and thus achieves better performance.
>
> * W3. The introduction of EDL in Section 3.2 is insufficient
>
> We have revised Section 3.2 to add more details in introducing the method.
>
> * W4. Missing discussion of visual localization (InLoc and AachenDay-Night) and homography estimation (HPatches)
>
> We added additional benchmarking datasets in A.3 of the revised paper. Our method obtains 1.7% increase in mAA@10px than RoMa on WxBS, and 0.5% increase in AUC$@3$px on HPatches. For visual localization (InLoc and AachenDay-Night), RoMa does not release their evaluation code. We need more time to reproduce their results and evaluate our method, and thus do not report the results in current version of the paper.
>
> * Q1. Difference or relationship of the proposed method compared to outlier filtering post-processing method like RANSAC
>
> The outliers in RANSAC refer to points that do not fit into the current estimation of the geometry model (e.g., homography, essential matrix). The certainty estimation in our task does not involve a geometry model. Instead, the matches sampled based on the estimated certainty will be used in downstream geometry estimation tasks, where RANSAC will be applied.

---

> ### Comment · Reviewer_Uwz6 · 2024-12-02
> **Thanks for the rebuttal**
>
> Thanks for the response. The rebuttal addressed most of my concerns, so I raised my score to 5. The remaining concern is that the novelty of this work is somehow incremental, i.e., incorporating EDL to dense feature matching intuitively. Moreover, the overall improvement of this work is mainly in corrupted images, which is not significant enough for the foundational feature-matching task.

---

### Official Review · Reviewer_oVDg · 2024-10-30

**Soundness:** 3
**Presentation:** 3
**Contribution:** 2
**Rating:** 8
**Confidence:** 4

**Summary:**

The authors propose modelling certainty of correspondences in dense matchers using an evidential deep learning approach. Instead of just estimating a certainty between 0 and 1, the model outputs the parameters of a Dirichlet distribution over probabilities of the two classes "the predicted correspondence is reliable" and "the predicted correspondence is unreliable". The certainty output is then the expected probability of the first class according to this Dirichlet distribution. The authors show experimentally by retraining the dense matcher RoMa that their approach leads to improved certainty scores, in particular leading to increased robustness to image corruptions.

**Strengths:**

1. The paper proposes a simple tweak to the RoMa-method, which gives good results experimentally.
2. Within the framework of the RoMa matcher, certainties are updated iteratively in each warp refinement step by a "logit-offset" in the original model. It seems more intuitive to let each refinement step produce positive evidence values for the two classes "correct" and "incorrect" that are summed over the steps, as is done in this paper. Perhaps, the authors could expand on this in the paper.
3. In general, outputting good certainties is an underexplored part of deep learning for 3D vision. The application of evidence based learning to dense matchers is novel.

**Weaknesses:**

1. The experiments are limited in that only a single model is tested. Hence, it is difficult to say if the improvements generalize to other models.
2. The explanation of evidential deep learning was difficult to understand, and I had to refer back to the original paper by Sensoy et al. I think this section could be improved.
3. Evidential deep learning has a built-in uncertainty measure. In the context of the present paper, we get a Dirichlet distribution over the classes "the predicted correspondence is reliable" and "the predicted correspondence is unreliable", and the associated uncertainty describes how spread out this Dirichlet distribution is. This uncertainty is however not used in the present approach. This makes it a bit difficult to interpret the method. We get a Dirichlet distribution but only use its expected value over the first class. This expected value should signify correspondence reliability, but there is also an uncertainty of this prediction inherent in the Dirichlet distribution, which is not used. Since RoMa uses regression-by-classification in the coarse matching step, a more natural approach may be to reformulate the loss for that classification over $N\times N$ image patches as evidential and use the uncertainty of the predicted Dirichlet distribution as an uncertainty score.


### Post-rebuttal:
- The authors have added experiments with DKM to address weakness 1.
- Weakness 2 has been addressed.
- Weakness 3 has not been addressed but is left for future work.
- My questions below have also been answered satisfactorily.

Looking at the other reviews there are two main remaining weaknesses
- The novelty is quite limited.
- The results are not significantly better than RoMa, except under major image degradations.

All in all, I will raise my score to "accept", but note the two weaknesses above.

**Questions:**

1. How much does the threshold used for balanced sampling matter for the robustness under image degradations? Both in the original RoMa and the new model.
2. My reading is that computationally, the new certainty estimation is more or less as heavy as the old in terms of inference and training speed. Is this correct?
3. Is the increased performance due to only better certainties? For example, is the performance the same if certainty scores from the new model are combined with matches from the original RoMa?
4. Is there a way to make use of the built in uncertainty in the Dirichlet distribution as described in Weakness 3?

---

> ### Author Response · Authors · 2024-11-28
>
> * W1. Only a single model is tested
>
> We verify our method on another dense matcher, DKM [Edstedt et al., 2023] in A.4 of the revised paper. Our method increases the clean AUC$@5 by 1.6%  on MegaDepth-1500, and increases the mean corruption AUC@5 by 1.3% on MegaDepth-1500-C. These results demonstrate the generalizability of our method beyond RoMa.
>
> * W2. Explanation of EDL is difficult to understand
>
> We have revised Section 3.2 to make it more clear.
>
> * W3. EDL's built-in uncertainty measure is not used and apply EDL on the regression-by-classification in the coarse matching step
>
> In our current approach, EDL's built-in uncertainty measure is not used, and instead its expected value over the first class is used to indicate matching reliability. Actually in the seminal work [Sensoy et al.,  NeurIPS2018], EDL's built-in uncertainty measure is also not used: for fair comparison with other methods, the authors choose to use the entropy computed from the expected class probabilities and demonstrate improved uncertainty measurement for out-of-distribution detection. This suggests that the expected class probability of EDL itself is informative and can be directly used for following tasks.
>
> It is possible to apply EDL for the regression-by-classification in the coarse matching step, but there are two drawbacks for this approach. First, the uncertainty at coarse scale does not necessarily represent the uncertainty at the finest scale. In the current method, only the matches and certainty score at the finest scale are used for balanced sampling and downstream tasks. As shown in the ablation study on coarse vs. fine scale EDL in the revised paper (Table 5), it is the EDL at fine scale that contributes most to the improvement, and coarse-scale EDL is not effective. Second, while RoMa use a regression-by-classification formulation in the coarse matching step, other dense matchers, e.g., DKM [Edstedt et al., 2023] and DGC-Net [Melekhov1 et al., 2019], use L1 or L2 regression for coarse matching. Nevertheless, all these dense matchers use the same classification formulation for certainty estimation. Our current formulation of EDL for certainty estimation thus avoids model-specific design and can be easily applied to other dense matchers.
>
> * Q1. How much does the threshold used for balanced sampling matter for the robustness?
>
> We observe that inappropriate choice of the threshold can adversely affect both RoMa and our method. For example, increasing the threshold from 0.05 to 0.1 degrades the performance of our method by 2.6%, and RoMa by 1.7% on MegaDepth-1500 corrupted by Gaussian noise at severity level 5. A sampling technique without such hardcoded threshold would be an interesting future work.
>
> * Q2. Is  the new certainty estimation  more or less as heavy as the old in terms of inference and training speed?
>
> Yes. Your reading is correct. We report the training and inference time of RoMa and our method in Table 6 of the revised paper. For training, our method takes 1.4% more GPU hours than RoMa. For inference, our method actually incur marginally lower cost than RoMa due to the elimination of sigmoid computation for obtaining final probability.
>
> * Q3. Is the increased performance due to only better certainties? Is the performance the same if certainty scores from the new model are combined with matches from the original RoMa?
>
> We added in A.5 of the revised paper the results of combining certainty score from our model with the warp from the original RoMa model. We observe that using our certainty score, the performance of RoMa is increased from 25.3% to 36.5%, close to our results of 37.9%. In Section 4.5 of the paper, we report the average endpoint error (AEPE), which is defined as the average Euclidean distance between the estimated and ground truth warp. The AEPE for our method and RoMa on MegaDepth-1500-C is 6.12 and 6.07, respectively. The comparable AEPE values suggest that the warp prediction branch of our method performs similarly to RoMa. These two studies combined suggest that it is mainly the better certainty estimation that brings the improvement in performance.
>
> * Q4. Is there a way to make use of the built-in uncertainty in the Dirichlet distribution?
>
> One way to utilize the built-in uncertainty may be to use it as a weighting on top of the predicted reliability score, so that matches with high reliability score and low uncertainty score are more likely to be sampled in the following balanced sampling step. However, our preliminary study shows that this weighting strategy does not produce better results. How the built-in uncertainty in EDL can be used to improve feature matching performance is an interesting and relevant problem and we would like to explore it in the future.

---

> > ### Comment · Reviewer_oVDg · 2024-12-02
> >
> > I thank the authors for the well-written rebuttal.
> >
> > I have one follow-up question regarding the comment
> > > We observe that inappropriate choice of the threshold can adversely affect both RoMa and our method. For example, increasing the threshold from 0.05 to 0.1 degrades the performance of our method by 2.6%, and RoMa by 1.7% on MegaDepth-1500 corrupted by Gaussian noise at severity level 5. A sampling technique without such hardcoded threshold would be an interesting future work.
> >
> > This sounds like reducing the threshold to (say) 0.025 could be a good idea. Or is 0.05 perfectly tuned for image corruptions?

---

> > > ### Author Response · Authors · 2024-12-02
> > >
> > > We would like to thank the reviewer for reading our rebuttal. For the follow-up question regarding the score threshold, we would like to clarify that 0.05 is the value used by DKM and RoMa in their paper and code. We do not tune this parameter but just follow their setting for fair comparison. We believe the authors of DKM and RoMa have carefully chosen this value based on their extensive benchmarking experiments. Reducing the threshold to 0.025 may work for  MegaDepth-1500, but may degrade on other benchmarks. A sampling strategy that avoids such hardcoded value should be a better solution.

---

> > > > ### Comment · Reviewer_oVDg · 2024-12-02
> > > >
> > > > I understand that 0.05 is used in prior work. However, it is not chosen for optimal performance on corrupted images. The presented method has an advantage in that it seems to work for uncorrupted and corrupted images with the same threshold, but it is still relevant to know if RoMa can be made to work on corrupted images simply by changing the score threshold.
> > > >
> > > > > A sampling strategy that avoids such hardcoded value should be a better solution.
> > > >
> > > > I agree, but my understanding is that the proposed method still needs a hardcoded balanced sampling threshold (the same value as in RoMa). Is this understanding incorrect?

---

> > > > > ### Author Response · Authors · 2024-12-04
> > > > >
> > > > > Dear Reviewer oVDg,
> > > > >
> > > > > Thanks for your further comments. We ran the experiment suggested by the reviewer. We observe that reducing the score to 0.025 improves RoMa's performance by 2.5% on MegaDepth-1500 corrupted by Gaussian noise at severity level 5. The best result 31.1% is obtained when we further reduce the score threshold to 0.01. Compared to our result of 37.1%, there is still a gap of 6%. This suggests that changing the threshold can only partially address the problem. We agree with the reviewer that it is relevant to know whether RoMa can be made to work on corrupted images simply by changing the score threshold. We will add these additional results in our revision.
> > > > >
> > > > > Yes. Your understanding is correct. We still use the same balanced sampling algorithm as in RoMa.

---

### Official Review · Reviewer_3kpK · 2024-10-31

**Soundness:** 3
**Presentation:** 4
**Contribution:** 3
**Rating:** 6
**Confidence:** 3

**Summary:**

This paper applies evidential deep learning to feature matching tasks, introducing an evidential learning framework for certainty estimation in dense feature matching problems. The proposed method enhances robustness in dense matching against corruptions and adversarial attacks, with extensive experiments conducted and visualization presented to demonstrate its performance.

**Strengths:**

1. The paper is well-motivated, and the main idea is clearly explained.

2. Experiments are conducted across a wide range of benchmarks with various types of corruptions and adversarial attacks. The proposed method outperforms in most cases.

3. The paper includes visualizations to analyze why the proposed method performs better than comparison method, particularly on corrupted data across different datasets.

**Weaknesses:**

Several questions need to be addressed:

1. This work employs a two-dimensional evidential deep learning (EDL) framework to certainty estimation in both coarse-scale and fine-scale losses. What would happen if EDL were applied to only one of these loss scales? Conducting an ablation study could provide insights into the effectiveness of EDL at each scale. It would be great if the authors could report performance results by applying EDL exclusively to coarse-scale or fine-scale losses, compared to using it on both losses.

2. Experiments are conducted on two datasets, MegaDepth-1500 and ScanNet-1500. There are other datasets mentioned in RoMa paper such as the street-view IMC 2022 and the challenging WxBS Benchmark. Evaluating the proposed method on those different datasets could further demonstrate its generalizability across diverse scenarios.

3. The proposed framework incorporates evidential deep learning into the training process. Could you provide details on how the proposed framework affects computational time, specifically in terms of training and inference times compared to the baseline RoMa method?

**Questions:**

Please refer to the weaknesses section.

---

> ### Author Response · Authors · 2024-11-28
>
> * W1. Ablation study on coarse-scale and fine-scale EDL
>
> We added an ablation study on applying EDL to different scales in Table 5 of the revised paper. We observe that applying EDL on the coarse scale alone is not effective, as the final prediction comes from the finest scale (which is still learnt by the BCE loss). Applying EDL on fine scales improve the performance on corrupted samples significantly. The best performance is achieved when EDL is applied on both coarse and fine scales.
>
> * W2. Experiments on IMC2022 and WxBS
>
> We added additional benchmarking results in A.3 of the revised paper. Compared to RoMa, our method obtains 1.7% increase in mAA@10px on WxBS, and 0.5% increase in  AUC@3px on HPatches. For IMC2022, RoMa does not release their evaluation code. We need more time to reproduce their results and evaluate our method, and thus do not report the results in current version of the paper.
>
> * W3. Computational cost of our method
>
> We report the training and inference time of RoMa and our method in Table 6 of the revision . For training, our method takes 1.4% more GPU hours than RoMa (128.4 vs. 126.6 GPU hours). For inference, our method actually incurs marginally lower cost (327 vs. 329 ms). This is probably due to the fact that EDL uses simple operations like addition and division to obtain the final probability, eliminating the more complicated sigmoid computation in RoMa.

---

> ### Comment · Reviewer_3kpK · 2024-12-01
>
> Thanks for the response. The authors presented results from the ablation study, experiments on the mentioned datasets as well as details on training and inference times. I think my concerns have been addressed and I will retain my positive rating.

---

### Official Review · Reviewer_vn4d · 2024-11-03

**Soundness:** 3
**Presentation:** 2
**Contribution:** 3
**Rating:** 5
**Confidence:** 3

**Summary:**

This paper presents an interesting idea of evidential learning for certainty estimation for the dense pixel-level feature matching task.
The proposed method is supposedly more OOD and adversarial robust than the current SotA RoMa.
It is tested against 2D Common Corruptions variants of 2 commonly used datasets for this task i.e. MegaDepth-1500 and ScanNet-1500.
It is also tested against outdated adversarial attacks such as FGSM and PGD.

**Strengths:**

If the idea of using evidential learning for feature matching is truly novel then that makes the work quite interesting and significant.
Apart from a couple of small typos, the paper is very well written.
The structure of the paper and the intended story are easy to follow.
The abstract of the paper is well written and to the point.

**Weaknesses:**

W1- **A lot of implementation details are missing from the paper.**
Simply mentioning that it is built on top of RoMa is insufficient information.
It is understandable to do so for the main paper to save space; however, the supplementary material should be used to provide such information, for example, the exact architecture, the training procedure, details about the datasets, HPC resources used, and other details important for reproducibility.

W2- **Needs a stronger argument for why OOD and Adversarial Robustness is important.**
The argument made in the introduction to explain why OOD and Adversarial robustness are important for this task can be made significantly stronger. Unfortunately, a case has not been made for why this is interesting and important for the community.

W3- **Out-dated evaluations for robustness.**
If the argument for OOD and Adversarial robustness is readiness for the real world, then the evaluations used do not hold up to the argument. Since the 3D Common Corruptions [1] are more real-world common corruptions than the 2D Common Corruptions used in the paper. Additionally, FGSM and PGD attacks were used for evaluating adversarial robustness, however [2] showed in their work that these attacks, originally proposed image classification are inadequate for pixel-wise prediction tasks, such as the one used in this proposed work. This is because FGSM and PGD optimize the attack by increasing the aggregate loss and not the per-pixel loss, this can cause the attack to be highly localized making a non-robust method appear very robust as the mean performance would still be quite well over the rest of the image space. Thus, specialized pixel-wise attacks such as CosPGD are essential for truly evaluating the adversarial robustness of pixel-wise prediction tasks.

W4- **Using 2D Common Corruptions on other known datasets is not always a novel contribution.**
It is unclear if the contribution of the 2 supposed OOD Robustness evaluation datasets MegaDepth-1500-C and ScanNet-1500-C is merely using 2D Common Corruptions proposed for ImageNet-1k and CIFAR datasets but changing their resolutions and applying them to the respective iid datasets or if there is more to the story, for example, some unforeseen complications that needed to be handled? If not, then simply applying these corruptions to other datasets is not exactly a novel contribution, it is still an interesting study just not a "new contribution" as claimed in the bullet points in the introduction of the paper.

W5- **Almost Redundant Presentation of Results.**
Including both Table 1 and Figure 3 is redundant. I understand that Table 1 contains the mean values over the 5 severity levels while Figure 3 shows the values at each severity, however by using straight dashed lines of respective colors, with y = mean value for all x values the need for Table 1 is eliminated.




**References**

[1] Kar, Oğuzhan Fatih, et al. "3d common corruptions and data augmentation." Proceedings of the IEEE/CVF Conference on Computer Vision and Pattern Recognition. 2022.

[2] Agnihotri, Shashank, Steffen Jung, and Margret Keuper. "CosPGD: an efficient white-box adversarial attack for pixel-wise prediction tasks." Forty-first International Conference on Machine Learning. 2024.

**Questions:**

Following are the questions for which I would highly appreciate an answer, these questions have not impacted my current recommendation for this paper, however, the response might have a significant impact on my final recommendations.

Q1- **Unclear evaluation details for adversarial attacks used.**
The epsilon values used for attack are starting from 0.1, 0.2 going up to 1., here the attacks l-infinity norm bounded? If yes, then what is the valid image space? Is it [0, 1] or is it [0, 255], meaning when epsilon = 1, does this mean that the epsilon is actually 1/255 (meaning that the valid image space is [0, 255]), or is the value of epsilon actually 1, meaning the entire image is nothing but adversarial noise? In this case, the image would also look semantically different to the human eye meaning that it will no longer be a valid adversarial attack.
And if the epsilon value is in fact 1/255, then the drop in performance is too significant for a very small epsilon value indicating the method is not truly robust to adversarial attacks. Could you also please comment on this?

Q2- **The idea of using Evidential Learning for Pixel-Matching is not entirely novel.**
While the exact downstream task in [3] is different from the one explored by this proposed work, the core ideas for both seem unusually very similar, the key difference being the distributions used, while [3] used a Normal Inverse-Gamma (NIG) distribution, this work uses a Dirichlet distribution. Would you please further highlight the key differences between the two other than some task-related implementation details?


**References**

[3] Chen Wang, Xiang Wang, Jiawei Zhang, Liang Zhang, Xiao Bai, Xin Ning, Jun Zhou, Edwin Hancock,
Uncertainty estimation for stereo matching based on evidential deep learning,
Pattern Recognition, Volume 124, 2022,108498, ISSN 0031-3203, https://doi.org/10.1016/j.patcog.2021.108498. (https://www.sciencedirect.com/science/article/pii/S0031320321006749)

---

> ### Author Response · Authors · 2024-11-28
>
> * W1. Implementation details are missing
>
> We added the required details in A.1 of the revised paper.
>
> * W2. Needs stronger argument for why corruption and adversarial robustness is important for dense feature matching
>
> Feature matching models are typically trained with 3D supervision, i.e., ground truth matches are established by using 3D information including camera pose and depth. 3D datasets are more expensive to collect and are usually smaller in size compared to 2D datasets. For example, the MegaDepth dataset used in our paper contains 260k images, while the scale of modern 2D image datasets is usually in millions (ImageNet-1k:1.3M, ImageNet-21k: 14M, JFT: 303M). The relatively small amount of training data may limit the model robustness to testing data that differ from the training distribution. Previous benchmarking datasets like IMC2022 and WxBS focus on evaluating images captured in different conditions, e.g., viewpoints, timings (day vs. night, years apart) and illuminations. However, these benchmarks do not consider those corruptions that are likely to occur in real world, e.g., various kinds of imaging noise, blurring caused by camera motion, reduced visibility caused by adverse weather. Adversarial attacks serve as a worst-case analysis of model robustness. When deploying the feature matching model in safety-critical applications, it is essential to understand the lower bound of correctness. Adversarial robustness is thus investigated in our paper.  We have modified the Introduction section to better highlight the relevance of the proposed study.
>
> * W3. Outdated evaluations for robustness
>
> We added experiments on 3D Common Corruptions (3DCC) [Kar et al., 2022] and CosPGD [Agnihotri et al., 2024] in A.2 of the revised paper. For 3DCC, we evaluate on three corruption types (low light noise, ISO noise and color quantization). Our method demonstrates significant and consistent advantage over RoMa, achieving up to 9.4% increase in AUC@5 on MegaDepth-1500-3DCC. We also observe that for corruption types that require depth information for generation, e.g., motion blur and fog 3D, the generation code of 3DCC cannot work for the MegaDepth dataset, resulting in unrealistic corrupted images (some examples are provided in Fig.9). We need more time to look into the 3DCC code to adjust the parameters and thus do not provide the results for other corruption types in 3DCC. For CosPGD, our method achieves up to 2.8% gain over RoMa.
>
> * W4. Using 2D Common Corruptions on other known datasets is not always a novel contribution
>
> We have modified our contribution as ``We propose to evaluate the robustness of feature matching methods under common image corruptions and adversarial attacks, which has not been studied in previous work".
>
> * W5. Almost redundant presentation of results in Table 1 and Figure 3
>
> In Table 1, we not only provide the mean AUC for each corruption type, but also the clean AUC and the mean AUC for all corruption types, which cannot be achieved by drawing dashed lines in each subfigure of Figure 3. Also, we believe it is informative to provide a table summarizing the numerical results for each dataset. Therefore, we choose to keep Table 1.
>
> * Q1. Evaluation details for adversarial attacks
>
> The attacks are L-infinity norm bounded. The valid image space is [-2.1, 2.6] (the images are first scaled to [0,1] and then mean-std normalized by using ImageNet mean and std values). Attacks with epsilon=1 approximately modify pixel value magnitude by 20%, which is noticeable yet still semantically meaningful to human eyes. In practice, adversarial attacks are desired to be imperceptible so it is unlikely to use such large epsilon values. Here we experiment with [0, 1] to reveal the trend of degradation under different perturbation budgets. We have modified the description in Section 4.3 to provide the evaluation details to avoid confusion.
>
> * Q2. Novelty of using  evidential learning for pixel-matching task and difference form [Wang et al., 2022]
>
> The different distributions mentioned in the comment (Normal Inverse-Gamma (NIG) distribution in [Wang et al., 2022] vs. Dirichlet distribution in our work) stem from the different formulations of evidential learning: [Wang et al., 2022] use the deep evidential regression formulation [Amini et al., NeurIPS2020], while ours use the classic classification formulation [Sensoy et al.,  NeurIPS2018]. The nature of task (stereo matching in [Wang et al., 2022] and certainty estimation in our work) necessitate different choice of formulations. The formulation in [Wang et al., 2022] cannot be used for our task -- it assumes a Gaussian distribution of predicted variable, which does not hold for a classification task. Therefore, despite both works dealing with pixel-matching task, it is formulated in very different ways. Our formulation of using evidential learning for certainty estimation has not been explored in previous work, which constitutes a novel contribution.

---

> > ### Comment · Reviewer_vn4d · 2024-12-02
> >
> > Dear Authors,
> >
> > Thank you very much for your response and the changes to the submission.
> >
> > Most of my concerns in the original review have now been answered, however, unfortunately, one of the answers to my question raises a very big concern for me.
> >
> > The epsilon values used for attack evaluations are in {0.1, 0.2, 0.3, 0.4, 0.5, 0.6, 0.7, 0.8, 0.9, 1.0} when the valid unnormalized image space is [0, 1].  To the best of my knowledge, this is highly unusual and does not align with previous works on adversarial attacks.
> >
> > As rightly mentioned in the response "adversarial attacks are desired to be imperceptible". Therefore, when in an image space of [0, 1], and when $\ell_{\infty}$-norm bounded, the usually used epsilon values are {1/255, 2/255, 4/255, 8/255} i.e. $\approx$ {0.00392156862, 0.00784313725, 0.0156862745, 0.03137254901}. However, the lowest epsilon value considered here is already '0.1'. This is a very very high epsilon value! Adversarial attack evaluations and comparisons at such high epsilon values do not really signify much, and thus the evaluations need to be corrected.
> >
> > I would have liked to give this feedback significantly earlier in the discussion phase when revisions were possible, however, (understandably) this question has been answered only recently.
> >
> > In light of the existence of this major concern in its current form, I recommend rejecting the paper allowing for this concern to be addressed. However, I am open to further clarifications and discussions.
> >
> > Best Regards
> >
> > Reviewer vn4d

---

> > > ### Author Response · Authors · 2024-12-03
> > >
> > > Dear reviewer vn4d,
> > >
> > > For the adversarial attack experiment, we follow the setup in the seminal work for evidential deep learning  [Sensoy et al., NeurIPS2018] (https://proceedings.neurips.cc/paper/2018/file/a981f2b708044d6fb4a71a1463242520-Paper.pdf). In Fig.4 of the paper, the FGSM attack is implemented with epsilon value from 0.1 to 1 on the MNISTdataset, in Fig.5 of the paper, the FGSM attack is implemented with epsilon value from 0.05 to 0.4 on CIFAR5 dataset. They do not mention the valid image space in the paper. We check their official tensorflow implementation at https://muratsensoy.github.io/uncertainty.html, in [322] , the line "nimg = np.clip(a=nimg,a_min=0,a_max=1)" indicates they use valid image space [0,1] in the experiment. Therefore, it seems to us it is common practice in the evidential learning literature to use large epsilon value to attack the model.
> > >
> > > Previously we have experimented with small epsilon value 0.001 and 0.01 for FGSM and PGD attack (we do not plot the results in the paper as the label will be cluttered in x-axis.) We provide the results here in the table below:
> > >
> > > AUC@5 on MegaDepth-1500 under FGSM attack:
> > > epsilon | 0.001 | 0.01 |
> > > ---------|-------|-------|
> > > RoMa   | 60.2 | 56.6 |
> > > Ours   |61.7| 57.9|
> > >
> > > AUC@5 on MegaDepth-1500 under PGD attack:
> > > epsilon | 0.001 | 0.01 |
> > > ---------|-------|-------|
> > > RoMa   | 53.2 | 49.8 |
> > > Ours   |53.9| 50.5|
> > >
> > > We observe clear advantage of our method over RoMa under small epsilon values as well. Actually, given our method outperforms other methods consistently in large epsilon values from 0.1 to 1, we do not see why the trend should be reversed for small epsilon values.

---

> > > > ### Comment · Reviewer_vn4d · 2024-12-03
> > > >
> > > > Dear Authors,
> > > >
> > > > Thank you for the prompt response.
> > > >
> > > > Firstly, to address "Actually, given our method outperforms other methods consistently in large epsilon values from 0.1 to 1, we do not see why the trend should be reversed for small epsilon values." : Currently, the evaluations begin from a very high epsilon value of 0.1, and the gap between RoMa and the proposed method is almost negligible until epsilon=0.1, compared to performances of other methods under adversarial attacks. Evaluations using epsilon 0.1 and higher are almost meaningless, as at this point, it is not an adversarial attack anymore, the permissible perturbation budget is called epsilon since it is a minimal value, hence the use of the word and variable epsilon. However, the perturbation budget of 0.1 is not small.
> > > > Moreover, the claim in this paper is that the proposed method is "more robust" than RoMa. Had the claim been "as robust as RomA" the current evaluations would have been acceptable. However, if the claim of "more robust" is to be proved, meaningful comparisons need to be made. I will explain "meaningful comparisons" in my next point.
> > > >
> > > > Secondly, I see that the NeurIPS paper you cited also follows a similar regime. Unfortunately, that paper did not have reviewers to point this out; fortunately, this paper does. Adversarial attacks are a method to test the reliability of all Deep Learning based methods, use of these attacks is not limited to evidential learning-based methods.
> > > > Starting from the FGSM paper to PGD, APGD, AutoAttack, SegPGD, and CosPGD, all well-known $\ell_{infinity}$-norm bounded white-box adversarial attacks use $\epsilon \in [\frac{1}{255}, \frac{12}{255}]$ and never more, as beyond this the value it is not really $\epsilon$ anymore, the perturbations are too large and thus they are not an admissible adversarial attack as they are not meaningful attacks anymore.
> > > >
> > > > I hope I have been able to put across this concern. I would strongly recommend fixing the adversarial attack evaluations.
> > > >
> > > > Best Regards
> > > >
> > > > Reviewer vn4d

---

> > > > > ### Author Response · Authors · 2024-12-04
> > > > >
> > > > > Dear Reviewer vn4d,
> > > > >
> > > > > Thanks for your feedback. We would like to clarify that in our experiment, the attack is added to the ImageNet std-mean normalized image in [-2.1, 2.6], not the original image in [0, 1] (that is why we answer our valid image space is [-2.1, 2.6]. We implement it in this way as it is the  ImageNet std-mean normalized image that is feed into the forward and loss function in our code). Based on your comments, a reasonable choice of epsilon values would be [1/255, 2/255, 4/255, 8/255, 12/255] in the [0, 1] image space. When we apply it to our image space,  the epsilon values need to be scaled by the reciprocal of ImageNet std, i.e., 1/0.225$\approx$4.44 (the ImageNet std is [0.229, 0.224, 0.225]. Strictly speaking, the epsilon value needs to be scaled differently for the three channels. Here we just take the median value 0.225 for simplicity since the values are very close). The equivalent epsilon values in our image space would be: [1/255, 2/255, 4/255, 8/255, 12/255]*4.44=[0.01741176, 0.03482353, 0.06964706, 0.13929412, 0.20894118]. Therefore, our original comparison at epsilon 0.1 and 0.2 should still be meaningful, where noticeable gain of our method over RoMa can be observed, and thus support our claim that our method is more robust than RoMa.
> > > > >
> > > > > We managed to run the experiment on the new set of epsilon values for our method and RoMa. The results are reported below. Here we directly use the equivalent attack value in the [0, 1] space for brevity (while experiments are still done using the actual epsilon values in our image space).
> > > > >
> > > > > AUC@5 on MegaDepth-1500 under FGSM attack:
> > > > > |epsilon | 1/255 | 2/255 | 4/255 | 8/255 | 12/255 |
> > > > > |---------|---|---|---|---|----|
> > > > > |RoMa | 54.1  |  50.3  |44.9    |38.9   |34.5    |
> > > > > |Ours| 55.2| 51.1|46.7 |40.9 |37.4  |
> > > > >
> > > > > AUC@5 on MegaDepth-1500 under PGD attack:
> > > > > |epsilon | 1/255 | 2/255 | 4/255 | 8/255 | 12/255 |
> > > > > |---------|---|---|---|---|----|
> > > > > |RoMa | 54.3  | 48.5   | 34.7   | 19.1  |  14.8  |
> > > > > |Ours|55.4 |49.7 |37.2 | 22.5|  16.7|
> > > > >
> > > > > We observe consistent and up to 2.9% gain for FGSM and 3.4% for PGD. We believe such results demonstrate the improved robustness of our method over RoMa. We will update the full set of results in our revision. We hope this helps to address your concern.

---

### Author Response · Authors · 2024-11-28

We would like to thank all the reviewers for their insightful comments. We have uploaded a revised version of our paper. Specifically, the following changes have been made to address reviewers' concerns:
1. Modified the Introduction section to better motivate the use of evidential deep learning (EDL) in certainty estimation for dense feature matching
2. Modified Section 3.2 to provide more details in introducing EDL.
3. Added Fig.7 in Section 4.5 to provide more insight into why employing EDL improves the performance.
4. Added an ablation study on applying EDL to different scales in Section 4.5
5. Added a table for computational cost comparison in Section 4.5
6. Added A.1 to provide implementation details in model architecture, datasets and training procedure
7. Added A.2 to provide benchmarking results on 3D Common Corruptions and CosPGD
8. Added A.3 to provide  results on additional benchmarking datasets
9. Added A.4 to verify our method with other dense feature matching method
10. Added A.5 to provide results on combining certainty score of our method and warp estimation of RoMa

All the changes have been highlighted in blue in the revision. Responses to specific comments are provided under each reviewer's comment.

---

> ### Author Response · Authors · 2024-12-02
>
> Dear reviewers,
>
> As today is the last day you can post a message to us, may we check if our rebuttal has addressed your concerns? Is there any clarification needed? We appreciate you taking the time to read our revision and response. Please do not hesitate to let us know if you have any further concerns. We will try our best to address them.

---

### Meta-Review · Area_Chair_6zc6 · 2024-12-21

**Metareview:**

This submission proposes a method to increase the corruption robustness of dense feature matching. To this end, the paper proposes to modify the model's certainty prediction to predict the parameters of a Dirichlet distribution over probabilities of a correspondence being reliable or unreliable.
The proposed approach is simple and effective, the paper is well written and the method has been evaluated and proven to be beneficial on 2 commonly used datasets, MegaDepth-1500 and ScanNet-1500, with common corruptions and FGSM and PGD adversarial attacks. While two out of four reviewers give final scores of 5, they agree on the merit of the submission in terms of improving the robustness.

**Additional Comments On Reviewer Discussion:**

Two of the reviewers rate the paper with a score of 5 even after the rebuttal. Reviewer vn4d initially had several concerns that were addressed during the rebuttal. The last update in numbers by the authors in order to improve the evaluation using adversarial attacks could address the remaining concerns. The AC strongly encourages the authors to transfer these results also to the paper. Reviewer Uwz6 has raised the score from initially 3 to 5 during the rebuttal, pointing out that the improved corruption robustness of feature matching by the approach is not translated into an improvement on clean data. While this is true, I agree with the other reviewers that the improved robustness is in itself a valuable contribution.

---

### Decision · Program_Chairs · 2025-01-22

Accept (Poster)